# High Overexpression of *SiAAP9* Leads to Growth Inhibition and Protein Ectopic Localization in Transgenic *Arabidopsis*

**DOI:** 10.3390/ijms25115840

**Published:** 2024-05-27

**Authors:** Ru Meng, Zhipeng Li, Xueting Kang, Yujia Zhang, Yiru Wang, Yuchao Ma, Yanfeng Wu, Shuqi Dong, Xiaorui Li, Lulu Gao, Xiaoqian Chu, Guanghui Yang, Xiangyang Yuan, Jiagang Wang

**Affiliations:** 1College of Agriculture, Shanxi Agricultural University, Jinzhong 030801, China; 13099094095@163.com (R.M.); 13934180450@163.com (Z.L.); 16634254236@163.com (X.K.); zhangyujia202203@163.com (Y.Z.); wyrwyr2000@163.com (Y.W.); 17852407913@163.com (Y.M.); 18398921181@163.com (Y.W.); dong-s-q@163.com (S.D.); lixiaorui@sxau.edu.cn (X.L.); lulugao2013@126.com (L.G.); chuxiaoqian@sxau.edu.cn (X.C.); yanggh1991@163.com (G.Y.); 2State Key Laboratory of Sustainable Dryland Agriculture (in Preparation), Shanxi Agricultural University, Jinzhong 030801, China; 3Hou Ji Laboratory in Shanxi Province, Shanxi Agricultural University, Jinzhong 030801, China

**Keywords:** amino acid permeases, alternative splicing, nitrogen, amino acid, foxtail millet

## Abstract

Amino acid permeases (AAPs) transporters are crucial for the long-distance transport of amino acids in plants, from source to sink. While *Arabidopsis* and rice have been extensively studied, research on foxtail millet is limited. This study identified two transcripts of *SiAAP9*, both of which were induced by NO_3_^−^ and showed similar expression patterns. The overexpression of *SiAAP9L* and *SiAAP9S* in *Arabidopsis* inhibited plant growth and seed size, although SiAAP9 was found to transport more amino acids into seeds. Furthermore, *SiAAP9-OX* transgenic *Arabidopsis* showed increased tolerance to high concentrations of glutamate (Glu) and histidine (His). The high overexpression level of *SiAAP9* suggested its protein was not only located on the plasma membrane but potentially on other organelles, as well. Interestingly, sequence deletion reduced SiAAP9’s sensitivity to Brefeldin A (BFA), and SiAAP9 had ectopic localization on the endoplasmic reticulum (ER). Protoplast amino acid uptake experiments indicated that SiAAP9 enhanced Glu transport into foxtail millet cells. Overall, the two transcripts of *SiAAP9* have similar functions, but SiAAP9L shows a higher colocalization with BFA compartments compared to SiAAP9S. Our research identifies a potential candidate gene for enhancing the nutritional quality of foxtail millet through breeding.

## 1. Introduction

Amino acids play a critical role in the growth and development of plants [1,2], being primarily transported through the xylem and phloem [3,4]. The long-distance movement of amino acids depends on specific transporters on the plasma membrane for loading and unloading [4,5,6]. Recent studies have emphasized the significant contribution of amino acid transporters (AATs) in nitrogen distribution from source to sink [7]. Furthermore, AATs are involved in regulating of crop yield [8], crop nutritional quality [9], seed development [10], carbon and nitrogen metabolism [11] and stress resistance [12].

Biographical information reveals that the AAP family is categorized under a subfamily of the AAAP family within the AAT superfamily [13,14]. Extensive research has been conducted on AAPs in *Arabidopsis* and other crops. In *Arabidopsis*, AtAAP1 plays a role in controlling the entry of amino acids into developing embryos and promoting protein synthesis in seeds [15,16]. The *Ataap2* mutant, under different nitrogen conditions, shows increased nitrogen allocation to leaves, thereby improving their photosynthetic capacity [17]. Furthermore, AtAAP3 and AtAAP6 are essential for in transferring amino acids to sink organs during pathogen attacks [18]. In rice, transporters like OsAAP1, OsAAP4, and OsAAP15 have been shown to increase rice tillering and grain yield by regulating neutral amino acid distribution [19,20,21], while OsAAP3 and OsAAP5 play an opposite effect [22,23]. Recent studies have highlighted that using the CRISPR/Cas9 system to knock out *OsAAP6* and *OsAAP10* can reduce grain protein content (GPC) and improve eating and cooking quality (ECQ) [24]. Moreover, the overexpression of *OsAAP6* in rice plants increases the protein content of rice grains, especially under nitrogen rich conditions [9]. Similarly, ZmAAP6 has also been identified as a crucial positive regulator of maize protein content and nutritional quality [25]. *GmAAP6a* transgenic soybean exhibits enhanced nitrogen transport to seeds, resulting in higher biological yield under low-nitrogen conditions [26]. On the contrary, the *StAAP1* antisense line inhibits amino acid transport from source organs to the phloem, leading to a significant reduction of amino acids in tubers [27]. In addition, Tian et al. also showed that TaAAP3, TaATLa2, and TaATLb13 are involved in the response to drought and high-temperature stresses by regulating amino acid distribution [12]. Research on foxtail millet AAPs has indicated that *SiAAP9* is prominently expressed in immature spikelets and seeds based on preliminary bioinformatics and RNA-seq analysis [28]. However, a detailed investigation into the functions of *SiAAP9* is lacking. Foxtail millet, a traditional crop in China, is known for its nutritional value [29], including higher protein and crude fiber content [30], as well as nine essential amino acids required by the human body [31]. Meanwhile, it also indicates that the consumption of foxtail millet can prevent diabetes to a certain extent [32,33].

In this study, we utilized *SiAAP9* T_3_ generation homozygous transgenic *Arabidopsis* material to investigate the biological functions of *SiAAP9L* and *SiAAP9S*. *SiAAP9S* lacks a large loop structure compared to *SiAAP9L* on Exon 4. *SiAAP9* is constitutively expressed and induced by NO_3_^−^. The overexpression of *SiAAP9* led to the suppression of the growth and development of transgenic *Arabidopsis* vegetative organs while significantly enhancing the amino acids in seeds. The transport of specific amino acid substrates by SiAAP9 was demonstrated by the tolerance of *SiAAP9-OX* transgenic *Arabidopsis* lines to high concentrations of Glu and His. Additionally, transformation experiments with foxtail millet protoplasts confirmed the ability of SiAAP9 to transport Glu into foxtail millet cells. An analysis of *SiAAP9* overexpression in *Arabidopsis* resulted in the ectopic localization of SiAAP9, while the study of the subcellular localization of SiAAP9 protein showed defective responses of the vesicle transport pathway, as sequence deletion reduced sensitivity to BFA, and excessive SiAAP9 protein colocalized with the ER. In conclusion, both *SiAAP9* transcripts hinder the growth and development of transgenic *Arabidopsis* while facilitating the absorption of amino acids.

## 2. Results

### 2.1. SiAAP9 Exhibits Two Alternative Splicing (AS) Events

Previous studies [28] have shown that *SiAAP9* may be highly expressed in immature spikelets and seeds during the filling stage. We supposed that *SiAAP9* may play an important role in regulating seed development. At first, *SiAAP9* was cloned using cDNA templates of Jingu 21 (Figure 1A). Surprisingly, the bands of the colony PCR that linked the *SiAAP9* coding sequence to the CV16 plasmid cloning vector were not on the same horizontal line (Figure 1B). Through sequence alignment, it was found that *SiAAP9* lacked a sequence on Exon 4 (Figure 1C and Appendix A). Therefore, we labeled the long sequence as *SiAAP9L* and the short sequence as *SiAAP9S*. Additionally, AAP transporters have previously been reported as plasma membrane transporters in *Arabidopsis* and rice [20,34], we predicted the SiAAP9 transmembrane domain and protein tertiary structure. Notably, we observed that the protein encoded by the long transcript exhibited an extra extensive loop structure spanning amino acids 200 to 250 in contrast to the protein encoded by the short transcript (Appendix A).

### 2.2. Expression Pattern Analysis of SiAAP9

To detect the complete, long, and short transcripts of *SiAAP9*, specific primers were designed (Figure 2A). An RT-qPCR analysis of different tissues of Jingu 21 showed that the expression levels of the two transcripts were higher in shoots and lower in leaves, flowers, and grains (Figure 2B–D), which is inconsistent with the RNA-seq data in previous studies. In addition, some studies have shown that the presence of multiple transcripts can affect RNA-seq results, as the presence of different transcripts can lead to differences in the expression levels of the same gene [35,36,37]. Therefore, we believe that the existence of two transcripts of *SiAAP9* is the reason for the inconsistency between RNA-seq results and RT-qPCR results.

In order to investigate whether *SiAAP9* is induced by short-term NO_3_^−^ treatment, we performed an RT-qPCR analysis and found that after 1 h of NO_3_^−^ addition, the expression level of *SiAAP9* two transcripts showed a significant increase, while there was no significant change after 1 h of NO_3_^−^ reduction (Figure 2E–J). This suggests that *SiAAP9* is induced by short-term NO_3_^−^ exposure. Additionally, transgenic *Arabidopsis* plants were used to observe the expression of *SiAAP9*. The results of GUS staining revealed that *SiAAP9* is expressed in all tissues throughout developmental stages, including seedlings (Appendix A), leaves (Appendix A), stomata (Appendix A), flower organs (Appendix A), pistils (Appendix A), roots (Appendix A), stamens (Appendix A), and developing seeds (Appendix A). However, *SiAAP9* showed low expression in the root tip (Appendix A).

### 2.3. Overexpressing of SiAAP9 Inhibits Growth and Development in Transgenic Arabidopsis

To further investigate the function of *SiAAP9*, we generated transgenic plants by overexpressing *SiAAP9* in *Arabidopsis* using the 35S promoter. Long and short transcripts each selected two independent T_3_ homozygous overexpressing lines, namely *LOX-1#*, *LOX-2#*, *SOX-1#*, and *SOX-2#*. Throughout the entire growth and development period of the transgenic plants, *LOX-1#* showed no significant difference compared to the WT in agronomic traits, and the other lines exhibited significant decreases in growth compared to the WT. The primary root length at 7 days of *LOX-1#* increased by 4.3%, while the *LOX-2#*, *SOX-1#*, and *SOX-2#* lines decreased by 5.0%, 5.9%, and 3.4%, respectively, compared to WT. There was no significant difference in the rosette leaves area at 3 weeks between *LOX-1#* and WT, while the *LOX-2#*, *SOX-1#*, and *SOX-2#* lines also significantly decreased by 56%, 16.8% and 20.1%, respectively. In addition, the plant height at 4–8 weeks obviously displayed dwarfism (Figure 3). Based on the research findings, it is hypothesized that the observed phenotypic variations could be attributed to the varying overexpression levels of *SiAAP9* across different lines. To investigate this, we employed RT-qPCR and Western Blot techniques to analyze the overexpression and protein levels of *SiAAP9*. The results demonstrated that the overexpression level of *SiAAP9* in *LOX-1#* was 100 times higher than that of WT, whereas the overexpression levels in the *LOX-2#*, *SOX-1#*, and *SOX-2#* lines were 1000–5000 times higher than that of WT (Appendix A). As expected, the variation in the overexpression levels of *SiAAP9* is a key factor contributing to the observed phenotypic differences in the *SiAAP9-OX* transgenic *Arabidopsis* lines. To verify the reliability of the results, we also measured the primary root lengths of *LOX-3#*, *LOX-4#*, *SOX-3#*, and *SOX-4#* at 14 d and found that the length of the primary root was correlated with the overexpression level, consistent with the above trend of change (Appendix A). Therefore, we can conclude that the extent to which the growth of transgenic *Arabidopsis* is inhibited by the overexpression of *SiAAP9* is dependent on the level of overexpression.

The size and weight of grains have a close relationship with crop yield and nutritional quality [38,39]. Additionally, the seed size and 1000-seed weight were also suppressed in *SiAAP9* transgenic plants compared to those in Col-0, in which *LOX-2#* has a more significant reduction compared to other lines (Figure 4A–D), indicating that overexpression of *SiAAP9* negatively regulates the size and weight of seeds. To further investigate whether overexpression of *SiAAP9* affects the content of amino acids in seeds, we measured the content of 20 amino acids in seeds and found that the amino acid content in seeds of *SiAAP9-OX* transgenic *Arabidopsis* lines was significantly higher than that in WT, except for Arg, Asn, and Asp. Notably, *LOX-2#* and *SOX-2#* have significantly higher contents of most amino acids than WT (Appendix A). These results indicate that *SiAAP9* has an inhibitory effect on the development of vegetative organs in *SiAAP9-OX* transgenic *Arabidopsis* lines, but increases the amino acid content in the seeds, which may increase the nutritional quality potentially.

### 2.4. Effect of SiAAP9 on Primary Root Growth Is Determined by Its Overexpression Level, Regardless of Nitrogen Availability

Previous studies have shown that under nitrogen-sufficient conditions, there is no significant difference in the primary root length between overexpressing *GmAAP6a* and WT. However, under nitrogen-free conditions, the primary root length of overexpressing GmAAP6a is significantly longer than WT [26]. Hence, we aim to explore the impact of varying nitrogen levels on the biological function of *SiAAP9*.

In this study, we investigated the response of *SiAAP9* to NO_3_^−^ and NO_3_^−^ free. When exposed to NO_3_^−^ free stress, *SiAAP9-OX* transgenic *Arabidopsis* lines have smaller aerial parts and fewer and shorter lateral roots in the primary root. Furthermore, after being cultured on a nitrogen rich medium for 4 days and then transferred to NO_3_^−^ free medium for 8 days, we observed that the primary root length, root variation length, and fresh weight of *LOX-1#* were slightly higher than those of WT, while the *LOX-2#* and *SOX-1#* lines were significantly reduced compared with WT. However, *SOX-2#* showed no significant changes (Figure 5). Interestingly, the seedlings transferred from 0 mM NO_3_^−^ medium for 4 days to 7 mM NO_3_^−^ medium for 8 days showed the same significant change trend (Appendix A). Combined with the research results in Figure 3 and Figure 4, both indicate that the low overexpression level of *SiAAP9* had no significant effect on the growth and development in *SiAAP9-OX* transgenic *Arabidopsis* lines, while the high overexpression level of *SiAAP9* had a significant inhibitory effect on them. Therefore, our results suggested that the growth and development of primary roots are mainly controlled by *SiAAP9* overexpression level, rather than long-term NO_3_^−^ regulation.

### 2.5. SiAAP9-OX Transgenic Arabidopsis Lines Exhibit Tolerance to High Concentrations of Glu and His

In order to clarify the amino acid substrates that SiAAP9 may transport, we conducted an investigation on the tolerance of *SiAAP9* transgenic *Arabidopsis* plants to different types of amino acids. By creating various amino acid concentration gradients, we observed phenotypic differences when exposed to 10 mM Glu and 5 mM His (Figure 6A,B); the results showed that *SiAAP9-OX* transgenic *Arabidopsis* lines had larger aboveground parts and longer roots compared to WT, while *SiAAP9-OX* transgenic *Arabidopsis* lines did not show significant tolerance to the other 17 amino acids (Appendix A). To eliminate the interference of inorganic nitrogen, we investigated the tolerance of *SiAAP9-OX* transgenic *Arabidopsis* lines to amino acids in medium with amino acids as the sole nitrogen source, we discovered that both the WT and *SiAAP9-OX* transgenic *Arabidopsis* lines wilted on a medium with Glu as the sole nitrogen source (Figure 6C,E,F). Conversely, on the medium with His as the sole nitrogen source, the root length and fresh weight of the *SiAAP9-OX* transgenic *Arabidopsis* lines significantly increased compared to WT (Figure 6D,G,H). These results indicated that *SiAAP9-OX* transgenic *Arabidopsis* lines are intolerant to high concentrations of Glu in the absence inorganic nitrogen while remaining tolerant to excessive His.

### 2.6. Two Proteins Encoded by SiAAP9L and SiAAP9S Have Different Subcellular Localization

To detect the subcellular localization of SiAAP9S and SiAAP9L and the effect of nitrogen on subcellular localization, 4-day-old seedlings cultured on 7 mM NO_3_^−^ or 0 mM NO_3_^−^ medium were used. It is worth noting that LOX-1# is only localized on the plasma membrane; other organelles are basically undetectable and can only be observed under specific conditions (Figure 7A). Compared to LOX-1#, SiAAP9 may localized on organelles in other lines (Figure 7B,E,F). Furthermore, by analyzing the fluorescence intensity of 40 cells, we observed that cells exposed to 0 mM NO_3_^−^ exhibited stronger fluorescence signals compared to those exposed to 7 mM NO_3_^−^ (Figure 7C,D,G,H). To further explore the specific localization of SiAAP9L and SiAAP9S, we utilized FM4-64 dye to locate on different organelles. The staining results indicate that LOX-1# has a high degree of plasma membrane localization, while other lines also have partial colocalization on TGN and PVC in addition to plasma membrane localization. Based on this observation, we hypothesize that a higher overexpression level of *SiAAP9* may disrupt the protein sorting process. It is known that BFA-sensitive transport pathways consist of the efflux pathway from the Golgi to the plasma membrane and the degradation pathway from the Golgi to the vacuole [40]. Surprisingly, after BFA treatment, we found that SiAAP9L exhibits significant colocalization with BFA compartment, while SiAAP9S does not. This suggests that sequence deletion weakens the sensitivity of SiAAP9 to BFA. We did not observe any significant effect of WM treatment that can cause the fusion of vacuolar precursors to become larger [41] (Appendix A). On the other hand, we speculate that SiAAP9 protein is detained on ER, and ER staining results indicate that both SiAAP9L and SiAAP9S are localized on ER (Appendix A). Overall, the high overexpression of *SiAAP9* alters the subcellular localization of its encoded protein, with differences in localization between long and short proteins.

### 2.7. SiAAP9 May Promote Glu Uptake in Jingu 21 Protoplast

In order to further validate the transport of Glu by SiAAP9, we conducted a protoplast amino acid uptake assay. Empty vector, *OE-SiAAP9L*, and *OE-SiAAP9S* transformed into protoplasts of Jingu 21 were cultured under dark conditions for 4 h using FITC labeled Glu. The results showed that the fluorescence intensity of protoplasts transformed with *OE-SiAAP9* was significantly higher than that of protoplasts transformed empty vector (Figure 8), indicating that SiAAP9 can transport more Glu into foxtail millet cells.

## 3. Discussion

### 3.1. Higher Concentration of Amino Acids Transported to Seeds by SiAAP9 Transporter May Inhibit the Growth and Development of Arabidopsis Shoots and Roots

Amino acid content is a crucial determinant of the nutritional quality of crop grains [42]. This study found that the content of amino acid levels in *SiAAP9-OX* transgenic *Arabidopsis* seeds was significant compared to that in the WT, potentially leading to the inhibition of *Arabidopsis* growth and seed size (Appendix A). Previous reports have shown that overexpression of *OsAAP3* in rice increased amino acid content in grains but hindered the growth of the second axillary bud and the number of filled grains per plant [22]. Similarly, the overexpression of *OsAAP5* in transgenic rice resulted in elevated levels of Lys, Arg, Val, and Ala in leaf sheaths and leaves while also inhibiting the growth of tiller buds and overall growth [23]. These findings align with the outcomes of the current study. However, the impacts of *OsAAP1* [20] and *OsAAP4* [19] exhibit contrasting effects. The amino acid composition of seeds in *SiAAP9* overexpression lines was analyzed, revealing Glu as the most abundant. This finding may account for the observed tolerance of *SiAAP9-OX* transgenic *Arabidopsis* lines to high Glu concentrations (Figure 6A). In addition, the transformation of foxtail millet protoplasts has further confirmed the higher concentration of Glu (Figure 8), supporting the conclusion that SiAAP9 may function as a specific transporter for Glu. As a signaling molecule in plants, glutamate plays an important role in regulating plant growth and enhancing resistance to adversity [43]. Therefore, the overexpression of *SiAAP9* may potentially reduce crop yield while simultaneously improving crop nutritional quality.

### 3.2. In the Absence of Inorganic Nitrogen, SiAAP9-OX Transgenic Arabidopsis Lines Were Intolerant to High Concentrations of Glu

In this study, it was observed that *SiAAP9-OX* transgenic *Arabidopsis* lines exhibited longer roots and higher fresh weight compared to the WT when grown in a medium containing inorganic nitrogen and Glu (Figure 6A). However, both *SiAAP9-OX* transgenic *Arabidopsis* lines and the WT showed signs of wilting when grown in a medium with Glu as the sole nitrogen source (Figure 6C). This suggests a potential interaction between inorganic and organic nitrogen, leading to the intolerance of *SiAAP9-OX* transgenic *Arabidopsis* lines to Glu high concentrations in the absence of inorganic nitrogen. Previous research has explored the impact of nitrate and Glu signaling pathways on the structure and growth of *Arabidopsis* primary roots. It has been observed that increases in exogenous L-Glu concentrations inhibit the growth of *Arabidopsis* primary roots while promoting lateral root branching [44]. Additionally, NRT1.1 has been found to counteract Glu’s inhibitory effects on the primary root growth by detecting nitrate signals at the primary root tip, thereby modulating primary root growth, development, and structure [45]. Building on these findings, it is hypothesized that SiAAP9, acting as inorganic nitrogen and amino acid transporter in plants, may sense nitrate signals to suppress the Glu signaling pathway, thereby reducing the tolerance of *SiAAP9-OX* transgenic *Arabidopsis* lines to high Glu concentrations. This could potentially enhance the ability of *OE-SiAAP9* to absorb more Glu when transformed into Jingu 21 protoplasts (Figure 8).

### 3.3. Two Transcripts of SiAAP9 Perform Identical Functions but Exhibit Different Subcellular Localizations

Alternative splicing is the process of generating different mRNA transcripts from a single gene, resulting in variations in the function and subcellular localization of these transcripts [46,47,48,49]. This phenomenon plays a crucial role in plant development and stress responses [50,51,52]. Previous studies have shown that the two transcripts of *TaNAK* have contrasting roles in regulating plant flowering time and morphology, along with distinct subcellular localizations [53]. In *Arabidopsis*, *PIN7a* and *PIN7b* exhibit nearly identical expression patterns and subcellular localization [54]. While *ScMYBAS1-2* and *ScMYBAS1-3* have opposite effects on crop yield regulation, both proteins are localized in the nucleus [55]. Under stressful conditions, AS can generate transcripts with similar functions but different regulatory properties. For instance, overexpressing both the long and short transcripts of *ZmPP2C26* in rice significantly reduces drought tolerance and impairs photosynthetic parameters [56]. In salt stress, the two AS transcripts of the ring-type E3 ligase *SRAS1* exhibit opposing responses [57], and *OsbZIP58β* demonstrates lower transcriptional activity compared to *OsbZIP58α* under high-temperature conditions [58]. Similarly, *TaGS3* produces five splice variants, with *TaGS3.5* significantly increasing grain weight upon overexpression, while *TaGS3.1* has the opposite effect [59]. The citrus *CiFD* gene has two transcripts, *CiFDα* and *CiFDβ*, involved in the flowering response to drought and low temperature, showing similar expression patterns across various tissues but differing in subcellular localization and transcriptional activity [60], aligning with the results of this experiment. Our study reveals consistent expression patterns of the two transcripts of *SiAAP9* in different tissues of Jingu 21, as well as in response to NO_3_^−^ induction and starvation conditions (Figure 2). Both transcripts were found to negatively impact plant growth and development, including root length, rosette leaf area, and plant height (Figure 3), as well as root growth and development under nitrate response and starvation phenotypes (Figure 5). Thus, we suggest that both *SiAAP9* transcripts have similar functions. Interestingly, SiAAP9S exhibited lower sensitivity to BFA compared to SiAAP9L (Figure 8), indicating the need for further investigation into the alternative splicing mechanisms of *SiAAP9L* and *SiAAP9S* transcripts.

### 3.4. High Overexpression of SiAAP9 May Lead to Ectopic Localization, Which Is Detrimental to Plant Growth and Development

In this study, LOX-1# exhibited normal plasma membrane localization (Figure 7A), which is consistent with previous findings of AAP transporters being localized on the plasma membrane and nuclear membrane in *Arabidopsis*, rice, cucumber, legumes, etc. [23,61,62,63,64]. However, LOX-2#, SOX-1#, and SOX-2# also have colocalization on various organelles (Figure 7B–D), reflecting that the high overexpression of *SiAAP9* results in ectopic protein localization. Furthermore, SiAAP9 was found to be collocated with the ER (Figure 8). The ER is involved in crucial cellular processes like protein synthesis, folding, assembly, transport, and lipid metabolism [65,66]. When encountering issues with protein folding and modification processes in the ER, the ER stress response is triggered [67,68]. Environmental factors, such as low temperature [69], high temperature [68], drought [70], and salt stress [71], can induce ER stress, prompting plants to adapt and survive under these conditions. Studies have shown that the overexpression of *TabZIP60s* in *Arabidopsis* activates ER stress-related gene expression, enhancing heat stress tolerance [72]. The overexpression of *NAC062* induces the expression of unfolded protein response (UPR) downstream genes in response to ER stress, helping alleviate ER stress [73]. The inhibition of *NAA50* has been linked to impeded plant growth and reduced photosynthetic capacity, leading to ER stress and the activation of defense pathways [74]. The results of subcellular localization were associated with the phenotypic analysis in Figure 3 and led us to the conclusion that *SiAAP9* can exhibit various biological functions as a result of its distinct localization. Therefore, we hypothesized that the ectopic localization of SiAAP9 could be a key factor contributing to the growth inhibition observed in transgenic *Arabidopsis*.

## 4. Materials and Methods

### 4.1. Plasmid Construction and Plant Transformation

The DNA sequence of *SiAAP9* (Seita.5G401000.1) and its encoded protein sequence were obtained from the Phytozome website (https://phytozome.jgi.doe.gov/pz/portal.html, accessed on 11 October 2021). The transmembrane domains of SiAAP9 were predicted using the web-based TMHMM Server v. 2.0 (http://www.cbs.dtu.dk/services/TMHMM/, accessed on 15 November 2021). To construct the *SiAAP9*-OE vector, a 1419 bp *SiAAP9* coding sequence was amplified using the primers 5′-CACGGGGGACGAGCTCGGTACCATGGACGTGGAGAAGATCGAGAGGA-3′ and 5′-GTGGCGCGCCGGGCCCTCTAGAGAGCTGCGTCTGGAAGATGGTG-3′ and inserted downstream of the *35S* promoter in p35S-MCS-EGFP-6×His-35S-Hyg (a binary vector) using KpnI and XbaI restriction enzymes. *SiAAP9* CDS sequence was cloned using 5′-TACCGTCGACGAGCTAAGCTTGATGGACGTGGAGAAGATCGAGAGGA-3′ and 5′-GGGAAATTCGAGCTCGGTACCTCAGAGCTGCGTCTGGAAGATGGTG-3′ primers and linked to the HindIII and KpnI restriction enzymes of binary vector pCUN-NHF from Aying Zhang laboratory [75] to construct a plasmid for transforming foxtail millet protoplasts, which are mainly used to study the transport of Glu. To analyze the expression pattern of *SiAAP9*, *SiAAP9 Promoter*::GUS vector was constructed, a 2737 bp *SiAAP9* promoter (upstream of the start codon ATG) was cloned using the primers 5′-TGTCAAACACTGATAGTTTAAACCTCGCTTGTGTATGCGTGCTTCG-3′ and 5′-CCCGGGGATCGATCCTCTAGACGCTGAAAGCTGTCTGTTGCAGCT-3′ and inserted into the CCDB position of *PAL47* (CCDB-GUSA-Hyg) vector with PmeⅠ and XbaI.

### 4.2. Plant Materials and Growth Conditions

*Arabidopsis thaliana* ecotype Columbia (Col-0) was used for all experiments and as the genetic background for corresponding plant transformations. To ensure the authenticity of the fusion constructs, a DNA sequence was performed prior to transformation into Agrobacterium strain GV3101 for the stable transformation of Arabidopsis. The Agrobacterium-mediated floral dipping method was employed for *Arabidopsis* transformation. Transformants were firstly selected on 1/2 MS medium supplemented with 1/2000 (50 mg/mL) hygromycin B (31282-04-9, Coolaber, Beijing, China), and then the selected positive seedlings were transferred to vermiculite to pass down generations. *LOX-1#*, *LOX-2#*, *SOX-1#*, and *SOX-2#* lines were detected for *SiAAP9* overexpression level through RT-qPCR and Western Blot analysis. T_3_ homozygous *SiAAP9-OX* transgenic *Arabidopsis* lines were used for phenotypic analysis. *Arabidopsis* plants were cultivated in a greenhouse under long-day conditions, with a photoperiod of 16 h of light and 8 h of darkness, at a temperature range of 18–22 °C. Conventional variety foxtail millet Jingu 21 seedlings cultured in 7 mM KNO_3_ or 0 mM KNO_3_ solution for 7 days were grown in 25–30 °C with 16 h of light and 8 h of darkness.

### 4.3. Histochemical GUS Staining Assay

For the tissue-specific expression assay, three independent *SiAAP9 pro:GUS Arabidopsis* lines were generated, various tissues of *SiAAP9 pro:GUS transgenic* lines were incubated in staining buffer (32 mM Na_2_HPO_4_, 18 mM KH_2_PO_4_, 5 mM K_3_Fe(CN)_6_, 5 mM K_4_Fe(CN)_6_, Triton X-100, and 1 mg/mL X-Gluc) at 37 °C overnight. After discoloring by incubation in a solution of 70% ethanol, the stained samples were observed using an Olympus BX51 microscope (MuSen, Shanghai, China).

### 4.4. RNA Isolation and RT-qPCR Analysis

The seeds of Jingu 21 were sterilized with 0.1% mercuric chloride solution for 10 min and were cultured in a bean sprout box with 7 mM KNO_3_ and 0 mM KNO_3_ for 7 days, respectively, changing the nutrient solution every three days. Then, 7-day seedlings were transferred to 0 mM KNO_3_ or 7 mM KNO_3_ solution for 10 min, 30 min, and 1 h, respectively, to explore whether *SiAAP9* is induced by NO_3_^−^. Total RNA was extracted using Trizol reagent according to the manufacturer’s instructions (ER501-01-V2, TransGen Biotech, Beijing, China). A total of 100 mg of seedlings was weighed out as a repeat, and RNA was extracted. First-strand cDNA was synthesized using 1 μg of total RNA extracted from each sample using reverse transcriptase (G592, ABM, Shanghai, China). RT-qPCR was performed in a 10 μL reaction volume containing 4 μL 2 × Realtime PCR Super mix (MF004-10, Mei5bio, Beijing, China), 1 μL cDNA solution, and 5 μL specific primers (2 μM) under the following conditions: 95 °C for 5 min (1 cycle), 95 °C for 20 s, 55 °C for 20 s, and 72 °C for 20 s (40 cycles), as well as 72 °C for 5 s (1 cycle) using real-time fluorescence quantitative PCR instrument (American Bole Bio-Rad CFX96). *Siactin2* (*Seita.8G043100*) was used as an internal control for normalization. Three biological and three technical replicates were analyzed for RT-qPCR experiments. The fluorescent quantitative primers for *SiAAP9*, *SiAAP9L*, *SiAAP9S*, and *Siactin2* are designed as shown in Appendix A.

### 4.5. Nitrogen and Amino Acid Treatment Assay

To analyze the response of WT and *SiAAP9-OX* transgenic *Arabidopsis* lines to KNO_3_ and KCl conditions, seeds were sterilized with 70% ethanol and grown on MGRL medium containing multiple trace elements; the concentration of NO_3_^−^ was set to 0 mM and 7 mM. Four-day-old seedlings grown on 7 mM KNO_3_ and 7 mM KCl media were transferred to 7 mM KCl and 7 mM KNO_3_ media for 8 days, respectively. The primary root length and root length variation of 20 seedlings were measured using Image J software (https://imagej.nih.gov/ij/, accessed on 28 September 2022), and the fresh weight was weighed. To investigate the tolerance of WT and SiAAP9-OX transgenic Arabidopsis lines to 20 common amino acids (tyrosine is soluble in HCl, so we did not study it). All amino acids were purchased from Coolaber, Beijing. Amino acids include L-glycine (Coolaber, 56-40-6), L-alanine (Coolaber, 56-41-7), L-valine (Coolaber, 72-18-4), L-leucine (Coolaber, 61-90-5), L-isoleucine (Coolaber, 73-32-5), L-methionine (Coolaber, 63-68-3), L-proline (Coolaber, 147-85-3), L-tryptophan (Coolaber, 73-22-3), L-serine (Coolaber, 56-45-1), L-cysteine (Coolaber, 52-90-4), L-phenylalanine (Coolaber, 63-91-2), L-asparagine (Coolaber, 70-47-3), L-glutamine (Coolaber, 56-85-9), L-threonine (Coolaber, 72-19-5), L-aspartic acid (Coolaber, 56-84-8), L-glutamic acid (Coolaber, 56-86-0), L-lysine monohydrate (Coolaber, 39665-12-8), L-arginine (Coolaber, 74-79-3), and L-histidine (Coolaber, 71-00-1). WT and *SiAAP9-OX* transgenic *Arabidopsis* lines were cultured on 1/2 MS medium containing different types and concentration gradients of amino acids for 12 days. We further explored the tolerance of *SiAAP9* transgenic *Arabidopsis* to amino acids by using amino acids with significant phenotypic differences as the sole nitrogen source to cultivate seedlings for 12 days.

### 4.6. Subcellular Localization under Different NO_3_^−^ Conditions

To explore the localization of SiAAP9 under 7 mM NO_3_^−^ and 0 mM NO_3_^−^, two transcript coding sequences were fused into vectors with a GFP tag driven by the *CaMV35S* promoter. Transgenic T_3_ generation *Arabidopsis* seeds were sterilized and incubated for four days and stained with amphiphilic styryl dye FM4-64 reagent, and the roots were photographed using a laser confocal microscope. Under different NO_3_^−^ conditions, Image J software (https://imagej.nih.gov/ij/, accessed on 6 May 2023) was used to calculate the fluorescence intensity of GFP in forty meristematic cells to determine the difference between 7 mM NO_3_^−^ and 0 mM NO_3_^−^ conditions.

### 4.7. Observation of SiAAP9 Localization by Reagent Treatment

FM4-64 (CD4673, Coolaber, Beijing, China) was used to stain plant materials while we observed cell contours and endocytic transport. They were stained with 4 μM FM4-64 for 1 min and located on the plasma membrane (PM). Then, they were stained for 5 min, and the seedlings were transferred to 1/2 MS medium for 15 min, 30 min, and 2–3 h, respectively, after which they were observed to be located on trans Golgi network/early endosome (TGN/EE), prevacuolar compartment (PVC), and vacuolar membranes. Two T_3_ lines of the two transcripts were stained separately, and laser confocal scanning was performed to observe the degree of colocalization. In addition to using FM4-64 staining to observe the location, an ER tracker (C1041S, Beyotime, Shanghai, China) was used to observe whether SiAAP9 was retained in the ER, which leads to the inability to move to the appropriate location to perform functions. For BFA treatment, 4-day seedlings were treated in liquid 1/2 MS medium plus 50 μM BFA (20350-15-6, Coolaber, Beijing, China) and 4 μM FM4-64 for 50 min. In addition, 33 μM WM (Wortmannin) (19545-26-7, Coolaber, Beijing, China) and 4 μM FM4-64 were added to 1/2 MS liquid medium for 1 h. The seedlings were washed three times and then were observed using a confocal laser scanning microscope.

### 4.8. Western Blot Analysis

Plant tissues were crushed after liquid nitrogen freezing and added to 100 μL 2 × sample buffer (0.1 mg/mL BPB, 40 mg/mL SDS, 20% (*v*/*v*) glycerol, 0.13 M tris-HCl). The sample was shaken thoroughly and then boiled at 90 °C for 3–5 min. After centrifugation at 12,000× *g* rpm for 10 min, proteins were separated using 8% (*w*/*v*) SDS-PAGE and transferred to a nitrocellulose (NC) membrane in a wet transfer system. A 1:5000 dilution primary antibody anti-GFP mouse mAb (D037, TransGen Biotech, Beijing, China), anti-actin (bsm-33128M, Bioss, Beijing, China), and 1:5000 dilution secondary antibody goat anti-mouse lgG, HRP (T156, TransGen Biotech, Beijing, China) were added to bind to specific proteins. After 2–3 h of incubation, the developer solution Supper ECL Immunoblotting Substrate (SL1350, Coolaber, Beijing, China) was added and the membranes were photographed with a protein imager (Vilber, Fusion FX6.EDGE, Marne-la-Vallee, France).

### 4.9. Protoplast Preparation and Transformation

The foxtail millet protoplast isolation method was modified according to the wheat and rice protoplast [76]. The shoot tissues of Jingu 21 seedlings that have grown for 7–14 days were cut with 0.5 mm to 0.4 M mannitol for 10 min in a dark condition. Then, plant materials were transferred to the 15 mL enzyme solution (20 mM MES (pH 5.7), 0.4 M mannitol, 20 mM KCl, 1.5% cellulase R10, 0.75% macerozyme, 10 mM CaC1_2_, 0.1% BSA), vacuumized for 30 min at 380~508 mm Hg, and immediately incubated for 5–6 h in the dark at 60–80 rpm. After enzymatic hydrolysis, W5 (154 mM NaCl, 125 mM CaC1_2_, 5 mM KC1, 2 mM MES) solution was added to release protoplasts. The supernatant obtained through filtration was centrifuged for 3 min at 250× *g* and then resuspended with an appropriate volume of W5. After a 30 min ice bath, sediment was suspended using MMG (0.4 M mannitol, 15 mM MgCl_2_, 4 mM MES). Finally, 10 μg of empty vector, *OE-SiAAP9L*, and *OE-SiAAP9S* were added to the foxtail millet protoplasts and incubated in a 24-hole plate for 16 h at 25 °C under dark conditions.

### 4.10. Protoplast Amino Acid Uptake Assay

Amino acids labeled with FITC (Glu-FITC) were synthesized by the Yuan Peptide Biotechnology Company, Nanjing, China. After dark incubation, the protoplasts were added to FITC-labeled Glu until reaching a final concentration of 0.1 mM in the dark for 4 h and centrifuged at 250× *g* for 3 min. The supernatant was carefully removed, and the protoplasts were suspended with 1 mL WI (4 mM MES, 0.5 M mannitol, 20 mM KCl) solution three times to remove free amino acids. Finally, the fluorescence signals were observed using a confocal laser scanning microscope.

### 4.11. Statistical Analysis

The data were analyzed using one-way ANOVA, followed by Student’s *t* and Duncan tests, with the following significance levels: * *p* < 0.05; ** *p* < 0.01; *** *p* < 0.001; **** *p* < 0.0001; or lowercase letters at *p* < 0.05.

## 5. Conclusions

In our study, we confirmed that the overexpression of *SiAAP9* inhibited the growth of vegetative organs in transgenic *Arabidopsis* but increased the concentration of most amino acids in the seeds. The protein encoded by a low overexpression level of *SiAAP9* is located in the normal plasma membrane, while the protein encoded by a high overexpression level is located in various organelles. Moreover, the ectopic localization of SiAAP9 protein on ER may also be one of the important reasons for inhibiting plants. The role of the alternative splicing mechanism of *SiAAP9* in plants and whether the colocalization of SiAAP9 and ER will result in ER stress phenomenon and hinder the growth of transgenic *Arabidopsis* plants need further research.

## Figures and Tables

**Figure 1 ijms-25-05840-f001:**
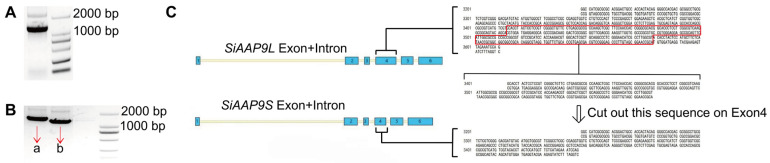
*SiAAP9* exhibits two alternative splicing events. (**A**) The agarose gel electrophoresis image displays the coding sequence of *SiAAP9*, which was cloned using the cDNA of Jingu 21 as a template. (**B**) Colony PCR agarose gel electrophoresis of *SiAAP9* encoding sequence linked to CV16 to construct cloning vector. (**C**) The schematic diagram illustrates the sequence alignment process after sequencing two colonies of a and b. The red box represents the sequence cut off by *SiAAP9* on Exon 4.

**Figure 2 ijms-25-05840-f002:**
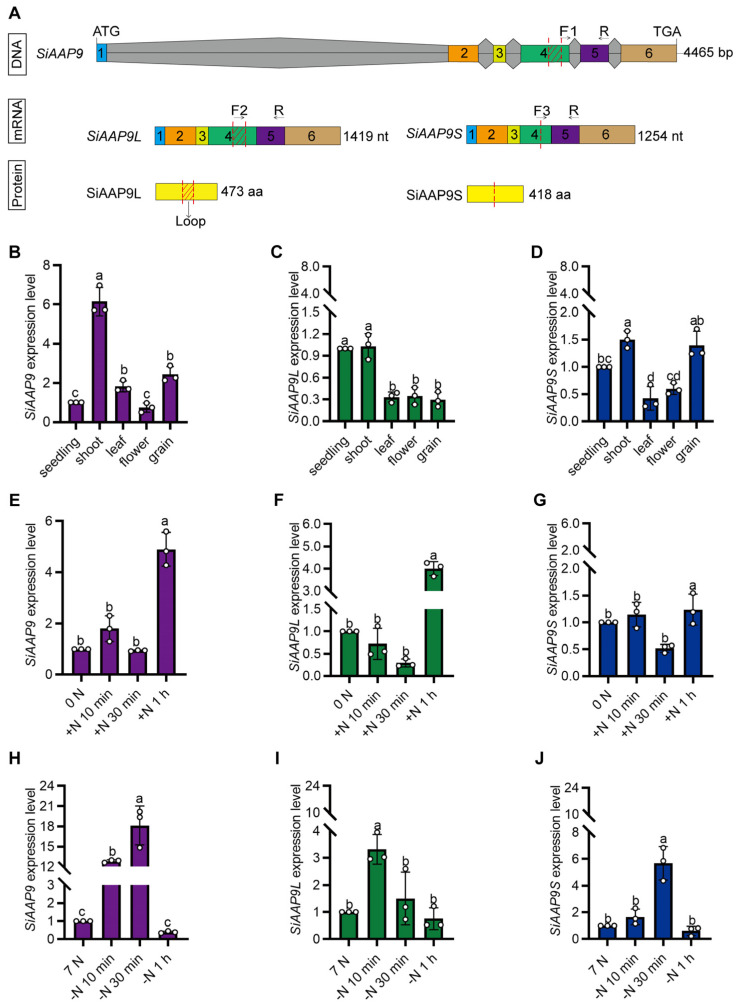
Analysis of *SiAAP9* expression patterns. (**A**) The schematic diagram illustrates the specific primer design for distinguishing between *SiAAP9*, *SiAAP9L*, and *SiAAP9S* transcripts. Among them, numbers represent exons, gray represents introns, other colors represent exons, arrows indicate the position of RT-qPCR primer design, and red dotted lines indicate the position of alternative splicing. (**B**–**D**) The expression levels of *SiAAP9*, *SiAAP9L*, and *SiAAP9S* transcripts in different tissues of Jingu 21. (**E**–**G**) The expression levels of *SiAAP9*, *SiAAP9L*, and *SiAAP9S* transcripts after adding NO_3_^−^ for 10 min, 30 min, and 1 h, respectively. (**H**–**J**) The expression levels of *SiAAP9*, *SiAAP9L*, and *SiAAP9S* transcripts after reducing NO_3_^−^ for 10 min, 30 min, and 1 h, respectively. (**B**–**J**) Significant difference is analyzed based on one-way ANOVA by Duncan test, lowercase letters at *p* < 0.05.

**Figure 3 ijms-25-05840-f003:**
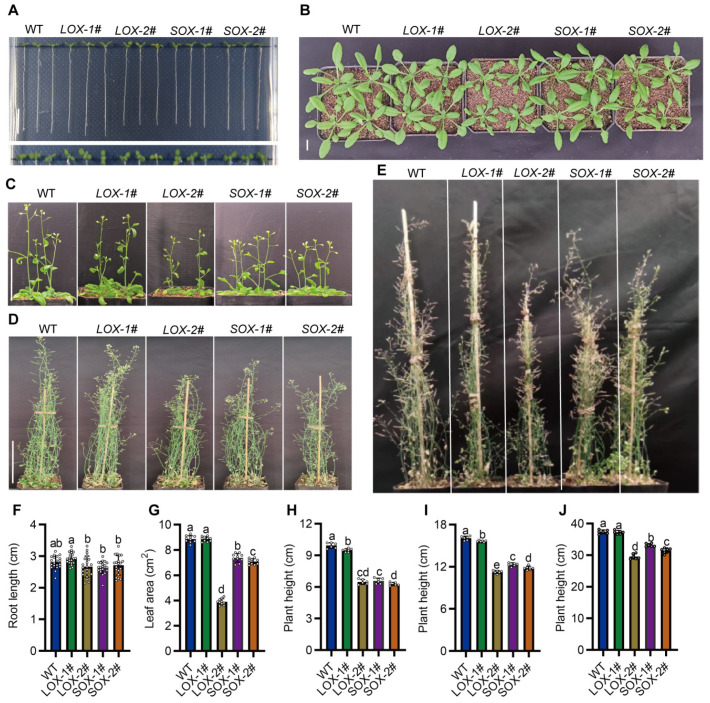
Phenotypic analysis of WT and *SiAAP9-OX* transgenic *Arabidopsis* lines under normal conditions. (**A**) Morphology of the seedlings at 7 days in WT and *SiAAP9-OX* transgenic *Arabidopsis* lines. Scale bar = 1 cm. (**B**) Morphology of the rosette leaves at 3 weeks in WT and *SiAAP9-OX* transgenic *Arabidopsis* lines. Scale bar = 1 cm. (**C**–**E**) Whole plant morphology of WT and *SiAAP9-OX* transgenic *Arabidopsis* lines at 4, 6, and 8 weeks. Scale bar = 5 cm. (**F**) Primary root length at 7 days was measured in WT and *SiAAP9-OX* transgenic *Arabidopsis* lines. Data represent means ± SD (*n* = 20). (**G**) Rosette leaf areas at 3 weeks were measured in WT and *SiAAP9-OX* transgenic *Arabidopsis* lines. Data represent means ± SD (*n* = 10). (**H**–**J**) Plant height at 4, 6, and 8 weeks were measured in WT and *SiAAP9-OX* transgenic *Arabidopsis* lines. Data represent means ± SD (*n* = 6). (**F**–**J**) Significant difference is analyzed based on one-way ANOVA by Duncan test, lowercase letters at *p* < 0.05.

**Figure 4 ijms-25-05840-f004:**
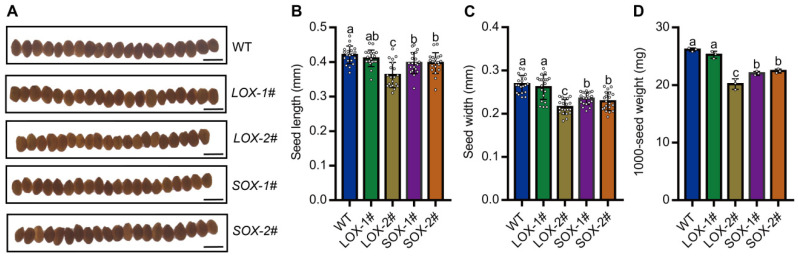
Overexpression of *SiAAP9* inhibited the seed size in *SiAAP9-OX* transgenic *Arabidopsis* lines. (**A**) Seed phenotypes of WT and *SiAAP9-OX* transgenic *Arabidopsis* lines after harvest. Scale bar = 500 μm. Comparison of the length (**B**), width (**C**), and 1000-seed weight (**D**) of seeds in WT and *SiAAP9-OX* transgenic *Arabidopsis* lines. Data represent means ± SD (*n* = 20 for seed size and *n* = 3 for seed weight). Significant difference is analyzed based on one-way ANOVA by Duncan test, lowercase letters at *p* < 0.05.

**Figure 5 ijms-25-05840-f005:**
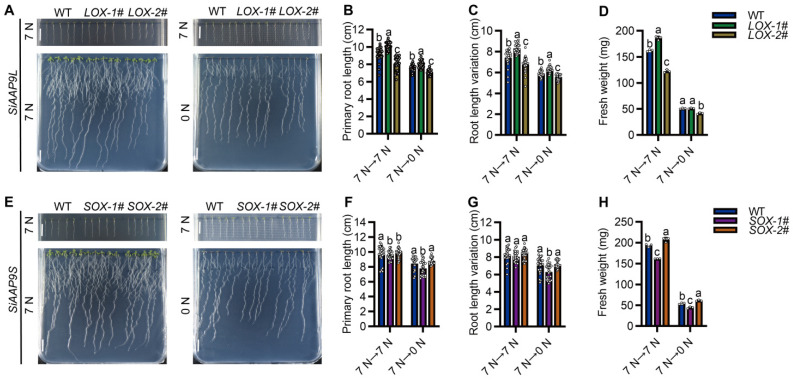
Phenotype analysis of WT and *SiAAP9-OX* transgenic *Arabidopsis* lines under NO_3_^−^ free stress. (**A**,**E**) WT, *LOX-1#*, *LOX-2#*, *SOX-1#*, and *SOX-2#* were vertically cultivated on 7 mM NO_3_^−^ medium for 4 days (upper panels), and then were transferred to 0 mM NO_3_^−^ medium and allowed to grow for an additional 8 days (lower panels). Scale bar = 1 cm. (**B**,**F**) Primary root length was measured in WT and *SiAAP9-OX* transgenic *Arabidopsis* lines at 12 days. Data represent the mean ± SD (*n* = 20). (**C**,**G**) Primary root length was measured in WT and *SiAAP9-OX* transgenic *Arabidopsis* lines after transferring the plate at 8 days. Data represent the mean ± SD (*n* = 20). (**D**,**H**) Fresh weight was measured in WT and *SiAAP9-OX* transgenic *Arabidopsis* lines at 12 days. Data represent the mean ± SD (*n* = 20). (**B**–**D**,**F**–**H**) Significant difference is analyzed based on one-way ANOVA by Duncan test, lowercase letters at *p* < 0.05.

**Figure 6 ijms-25-05840-f006:**
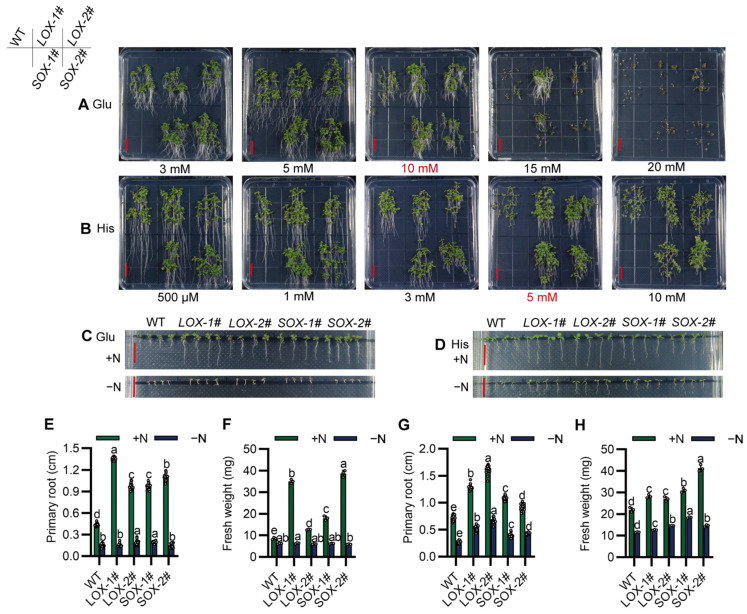
Overexpression of *SiAAP9* in *Arabidopsis* exhibited tolerance to high concentrations of Glu and His. (**A**,**B**) WT and four *SiAAP9-OX* transgenic *Arabidopsis* lines were grown for 12 days on 1/2 MS medium containing Glu or His. Scale bar = 1 cm. (**C**,**D**) WT and four *SiAAP9-OX* transgenic *Arabidopsis* lines were grown for 12 days on 1/2 MS medium with either normal N (+N) or 10 mM Glu (−N) or 5 mM His (−N) as the sole N source. Scale bar = 1 cm. (**E**,**G**) Primary root length was measured on 1/2 MS medium containing 10 mM Glu or 5 mM His and 10 mM Glu (−N) or 5 mM His (−N) as the sole nitrogen source. Data represent the mean ± SD (*n* = 20). (**F**,**H**) Fresh weight was measured on 1/2 MS medium containing 10 mM Glu or 5 mM His and 10 mM Glu (−N) or 5 mM His (−N) as the sole nitrogen source. Data represent the mean ± SD (*n* = 3). (**E**–**H**) Significant difference is analyzed based on one-way ANOVA by Duncan test, lowercase letters at *p* < 0.05.

**Figure 7 ijms-25-05840-f007:**
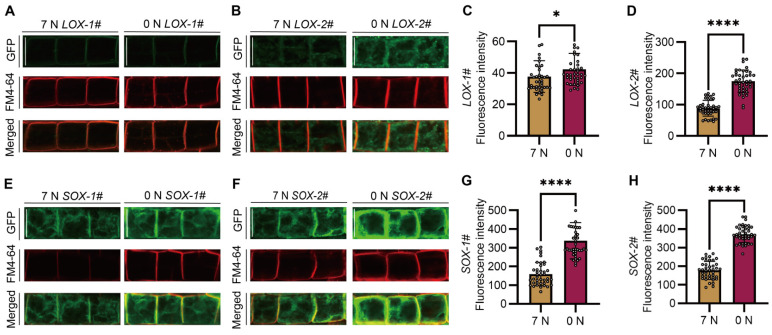
Subcellular localization analysis of SiAAP9L and SiAAP9S in *Arabidopsis* under two nitrate conditions. (**A**,**B**,**E**,**F**) Four *SiAAP9-OX* transgenic *Arabidopsis* lines were cultured vertically on 1/2 MS medium for 4 days, and meristematic zone cells of the roots were photographed using a confocal laser scanning microscope. Scale bar = 20 μm. (**C**,**D**,**G**,**H**) Fluorescence intensity of GFP signal of four *SiAAP9-OX* transgenic *Arabidopsis* lines was calculated under 7 mM nitrate and 0 mM nitrate treatment; each treatment was calculated for 40 cells, respectively. Data represent the mean ± SD (*n* = 40). Asterisks indicate significant differences based on one-way ANOVA followed by Student’s *t* test: * *p* <0.05, **** *p* < 0.0001.

**Figure 8 ijms-25-05840-f008:**
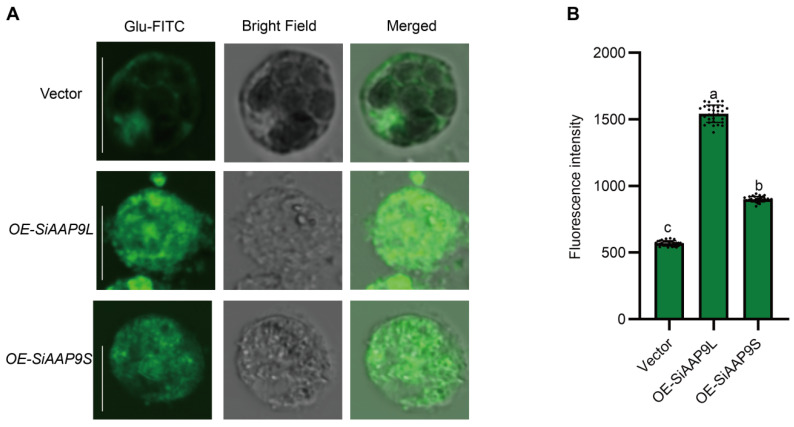
Amino acid uptake analysis in Jingu 21 protoplasts. (**A**) Green fluorescence images of protoplasts transformed by an empty vector, *OE-SiAAP9L*, and *OE-SiAAP9S* under Glu-FITC treatment. Scale bars = 10 μm. (**B**) Statistical analysis of cell fluorescence signal intensity. Fluorescence intensity of 30 cells was measured when maintaining consistent setting parameters. Data represent the mean ± SD. Significant difference is analyzed based on one-way ANOVA by Duncan test, lowercase letters at *p* < 0.05.

## Data Availability

The original contributions presented in this study are included in the article/Appendix A. Further inquiries can be directed to the corresponding author/s.

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
