# Peer review of "High Overexpression of SiAAP9 Leads to Growth Inhibition and Protein Ectopic Localization in Transgenic Arabidopsis"

_ijms, 2024, doi:10.3390/ijms25115840_

Round 1
Reviewer 1 Report
Comments and Suggestions for Authors
The Authors submit an article in which they report on their studies concerning the expression and localization of an amino acid transporter SiAAP9 in Arabidopsis. The study has a good premise in that amino acid transport is the basis of plant development and growth, and therefore its correct and regulated functionality is also the basis for obtaining crops in plants that produce food and give important economic returns. For this paper, one can see from the extensive bibliography that the authors have well researched and studied and adequately designed the experiments they conducted. There is a lot of work in this paper.
This is immediately apparent from the abstract and the introduction, although I would point out that in these parts of the paper an adequate revision of the English language is necessary (as well as in the discussion). This need is felt less in the technical part of the paper (results and materials and methods).
The introduction should have more linked sentences, it should give less the impression of a simple list of observations on transporters made by other researchers.
In any case, the work is well done, the results are thorough and the conclusions the authors draw are consistent with their observations. They also highlight some perspectives and spin-offs of their work, which perhaps they should beat about the bush even more, on the strength of their results.
The figures and supplementary material help a lot in reading and understanding the observations made, only that some of them are really small and, as a consequence, require not to convey their content to the reader (Fig. 4 A, Fig. 5 A and F, Fig. & A, B and C, Fig. 7 A, B, E and F)
Moderate editing of English language is required
Reviewer 2 Report
Comments and Suggestions for Authors
Dear Editor,
In the manuscript with the title “High expression of SiAAP9 leads to growth inhibition and protein ectopic localization in transgenic Arabidopsis” Meng et al studied the impact of the overexpression of the foxtail millet SiAAP9 amino acid permease in Arabidopsis development. Their findings showed that SiAAP9 inhibited plant growth and seed size as well as that SiAAP9 can transport more amino acids into seeds. The experimental work is well designed, performed and presented.
Please find bellow some comments or suggestions that can improve the manuscript according to my opinion.
1. In lines 79-81: “Overexpression of SiAAP9 resulted in ectopic localization of SiAAP9, and the vesicle transport pathway indicated that sequence deletion reduced sensitivity to BFA and excessive SiAAP9 protein colocalized with ER.”
Please consider: Analysis of SiAAP9 overexpression in Arabidopsis resulted in ectopic localization of SiAAP9, while the study of the subcellular localization of SiAAP9 protein showed defective responses of the vesicle transport pathway as……..
2. In figure 2: In the diagrams that show SiAAP9L and SiAAP9S expression levels (Fig2C,D, Fig2F,G and Fig.2I, J) make the scales in Y axis (minimum-maximum) uniform to make the comparison better visible.
3. In line 135 in the title of 2.3 paragraph you study the impact of SiAAP9 overexpression on Arabidopsis development. It is not just SiAAP9 expression.
4. In line 143: Please specify the “other three lines”
5. In lines 159-160: I am not sure that your conclusion is right. Generally, overexpression or ectopic expression of genes/proteins can cause developmental problems. Please reconsider your conclusion.
6. In the Figure 3 legend (line 161) you mention “growth phenotype analysis of…” I think it would be better “phenotypic analysis of.. or developmental analysis of…”
7. In the legend of Figure 4 (line 186) I think you should mention that overexpression of SiAAP9 inhibited…..
8. In line 211: I think you should find a better way to connect the main outcome that is described in the previous sentence with your conclusions. A possible way is : Therefore, our results suggest ….
9. In line 241 please consider replacing “after lacking” with “in the absence “
10. In figure 8A. The protoplasts of OE-SiAAP9L and OE-SiAAP9S seem to be dead. Please replace the presented images with better ones.
11. In line 310: Please remove “and” from the beginning of the sentence.
12. In line 316 please replace “But” with “However”
13. In line 317: I think you should add In SiAAP9 overexpressing lines among …
14. In line 320: “has been further confirmed “ should change to “has further confirmed the higher concentration of Glu”
15. In lines 322-324: Consider revising English
16. In lines 329-330: Consider revising English
17. In line 391: Remove And before Overexpression.
18. In line 396: Consider replacing “We associate” with the following “The results of subcellular localization associate with the phenotypic analysis in Figure 3 and led us to the conclusion that .....

I think that moderate English language editing is required
